# Non-Uniform Voxelisation for Point Cloud Compression

**DOI:** 10.3390/s25030865

**Published:** 2025-01-31

**Authors:** Bert Van hauwermeiren, Leon Denis, Adrian Munteanu

**Affiliations:** Department of Electronics and Informatics (ETRO), Vrije Universiteit Brussel, Pleinlaan 2, 1050 Brussels, Belgium; ldenis@etrovub.be

**Keywords:** point cloud, compression, quantisation, voxelisation

## Abstract

Point cloud compression is essential for the efficient storage and transmission of 3D data in various applications, such as virtual reality, autonomous driving, and 3D modelling. Most existing compression methods employ voxelisation, all of which uniformly partition 3D space into voxels for more efficient compression. However, uniform voxelisation may not capture the underlying geometry of complex scenes effectively. In this paper, we propose a novel non-uniform voxelisation technique for point cloud geometry compression. Our method adaptively adjusts voxel sizes based on local point density, preserving geometric details while enabling more accurate reconstructions. Through comprehensive experiments on the well-known benchmark datasets ScanNet, ModelNet and ShapeNet, we demonstrate that our approach achieves better compression ratios and reconstruction quality in comparison to traditional uniform voxelisation methods. The results highlight the potential of non-uniform voxelisation as a viable and effective alternative, offering improved performance for point cloud geometry compression in a wide range of real-world scenarios.

## 1. Introduction

In recent years, the demand for point cloud processing techniques has surged, driven by their increasing application in emerging fields such as augmented and virtual reality (AR/VR), autonomous driving, and robotics. These applications often rely on the storage and transmission of large point clouds, which typically consist of millions of individual points. Consequently, the development of efficient point cloud compression methods has become critical to addressing the substantial data demands inherent in these systems. Unlike the structured nature of 2D image and video data, 3D point clouds are inherently unstructured, presenting unique challenges in compression. This complexity highlights the need for specialised algorithms and has driven extensive research efforts aimed at optimising point cloud compression techniques.

Nearly all point cloud compression methods in the literature begin by voxelising the input point cloud, converting it into a grid of discrete, integer-coordinated voxels. This voxelisation process not only imposes structure on the otherwise unstructured point cloud but also makes the data more amenable to compression techniques, such as convolution-based approaches and octree partitioning schemes. By introducing a structured representation, voxelisation enables compression algorithms to more effectively exploit spatial redundancies and patterns within the data.

The most established point cloud compression codecs are being developed by the Moving Picture Experts Group (MPEG), the organisation responsible for widely adopted video coding standards such as AVC [1], HEVC [2], and VVC [3]. They are maintaining two distinct point cloud compression standards: the geometry-based point cloud compression standard (G-PCC) [4] and the video-based point cloud compression standard (V-PCC) [5]. The V-PCC standard leverages the power of established video codecs by projecting the 3D point cloud onto 2D planes and then compressing these projections as video streams. This approach benefits from the maturity and optimisation of existing video compression technologies.

The G-PCC standard, on the other hand, works purely in 3D and has two variants. The first option is an octree-based codec, which recursively splits the bounding box around the voxelised point cloud into eight subnodes until the nodes are voxel-sized; at this point, the octree leaves are all the occupied voxels. The only information to be encoded are the occupancy bits, which indicate which subnodes are occupied. So, most active research investigates how to efficiently entropy code these using predictions from recently coded or neighbouring nodes. The second approach, which performs predictive geometry coding, is designed for scenarios requiring low-complexity decoding and minimal latency, such as streaming applications. This method uses a prediction graph to model the point cloud, where each point’s geometry is estimated based on its predecessors in the graph. This predictive framework enables streamlined compression while maintaining efficient decoding.

Most point cloud compression research can also be classified into geometry-based [6] and video-based codecs [7,8]. The focus of this paper is on voxelisation, which is used in geometry-based codecs. Recent geometry-based point cloud compression research has increasingly focused on learning-based approaches, which can typically be grouped into four main categories. Voxel-based methods process voxelised point clouds directly using 3D convolutions, either in dense [9,10,11] or sparse forms [12,13,14], with the latter accounting for the inherent sparsity of point clouds. Octree-based methods build upon the octree structure used in the G-PCC standard but enhance it with machine learning to improve the entropy coding of occupancy information. For instance, VoxelContext [15] employs convolutional neural networks to predict occupancy probabilities, while transformer-based techniques such as Octattention [16], Octformer [17] and EHEM [18] model complex dependencies between octree nodes to boost compression efficiency. Hybrid methods combine elements of voxel-based and octree-based approaches. For instance, VoxelDNN [19,20] employs a pruned octree structure and encodes its leaves using a voxel-based approach. Finally, some methods operate directly on the unvoxelised version of the point cloud. Examples include approaches based on PointNet++ [21], such as those proposed in [22,23], or methods employing recurrent neural networks [24]. However, these techniques often struggle to scale effectively to higher bitrates and larger point clouds [12,25].

To the best of our knowledge, all existing point cloud compression methods operating on voxelised data rely on uniform voxelisation. However, this approach fails to consider the spatial distribution of points within the input data. As a result, the bitrate is often misallocated—excessively spent in sparse regions while insufficiently invested in dense, detail-rich areas. To address these shortcomings, we propose a novel non-uniform voxelisation method that adapts to the specific point distribution of the input cloud, optimising the allocation of resources to minimise Euclidean coding error. Our approach is compatible with any point cloud codec designed for voxelised data. By demonstrating the substantial benefits of non-uniform voxelisation compared to the traditional uniform approach, this work highlights the need to reconsider voxelisation strategies for advancing the state of point cloud compression.

We devise a novel approach that allows for non-uniform voxelisation in point cloud compression. The obtained method is based on Lloyd–Max quantisation and minimises Euclidean error.We empirically prove the potential of these approaches by a thorough testing on three well-known public datasets: ScanNet [26], ModelNet10 [27] and ShapeNet [28].

The remainder is structured as follows: first, the necessary basics of quantisation are explained in Section 2. Then, the proposed methodology is explained in detail in Section 3. The experimental setups and their results are summarised in Section 4. The strengths, limitations and future work are discussed in Section 5. Finally, conclusions are drawn in Section 6.

## 2. Background Knowledge

The voxelisation of a point cloud is quantising the positions of the points along the three spatial axes. Quantisation is the process of mapping a continuous range of values to a discrete set of symbols. Formally, a quantiser is characterised by its decision boundaries $b_{k=0}^{N}$, where $b_{0}=-\infty$ and $b_{N}=+\infty$, with consecutive pairs of boundaries defining bins that are associated with reconstruction values $y_{k=1}^{N}$. So, the quantisation function can be expressed as(1)$$Q\left(x\right)=y_{k}\Leftrightarrow b_{k-1}<x\leq b_{k}.$$

The goal of designing a quantiser is to minimise the bitrate while also minimising the distortion introduced in the process. This goal is often formalised by using a Lagrangian multiplier λ and minimising D+λR, where the distortion is usually given by the mean squared error(2)$$\begin{array}{cc} D & =E[{\left(x-Q\left(x\right)\right)}^{2}]\\ \; & =\int_{-\infty }^{\infty }{\left(x-Q\left(x\right)\right)}^{2}f\left(x\right)dx\\ \; & =\sum_{k=1}^{N}\int_{b_{k-1}}^{b_{k}}{\left(x-y_{k}\right)}^{2}f\left(x\right)dx. \end{array}$$

Meanwhile, the rate is given by(3)$$\begin{array}{cc} R & =\sum_{k=1}^{N}length\left(c_{k}\right)\int_{b_{k-1}}^{b_{k}}f\left(x\right)dx, \end{array}$$
where $c_{k}$ is the codeword that is assigned to quantisation bin *k*. For a sufficiently efficient entropy codec, the cost of each codeword approaches the information content $length\left(c_{k}\right)\approx -log\left(p_{k}\right)$, so the total rate approaches the entropy $H=-\sum_{k=1}^{N}p_{k}\log\left(p_{k}\right)$.

In this work, we look at two quantisers.

1.UniformThe uniform quantiser is popular due to its simplicity, and it is shown to be optimal for some distributions. Given a series of values in the range $\left[X_{min},X_{max}\right]$ the reconstruction values are given by(4)$$\begin{array}{c} y_{k}=X_{min}+\frac{2k-1}{2N}\left(X_{max}-X_{min}\right), \end{array}$$
where *N* is the number of quantisation levels. The decision boundaries are then in the middle of the adjacent reconstruction levels,(5)$$\begin{array}{c} b_{k}=\frac{y_{k}+y_{k+1}}{2}. \end{array}$$2.Lloyd–MaxThe Lloyd–Max quantiser adapts to the distribution of the input data f(x). When λ in the optimisation problem is set to 0, the problem becomes minimising the mean squared error, which can be solved:(6)$$\begin{array}{c} \frac{\partial D}{\partial b_{k}}=0\Rightarrow b_{k}=\frac{y_{k}+y_{k+1}}{2} \end{array}$$
and(7)$$\begin{array}{c} \frac{\partial D}{\partial y_{k}}=0\Rightarrow y_{k}=\frac{\int_{b_{k-1}}^{b_{k}}xf\left(x\right)dx}{\int_{b_{k-1}}^{b_{k}}f\left(x\right)dx}. \end{array}$$These formulas can be used in an iterative manner. We start with a uniform quantiser, then iteratively update the boundaries and reconstruction levels according to these rules until the average variance in the bins no longer decreases.

## 3. Proposed Methodology

We propose a novel non-uniform voxelisation method designed to minimise the square of the Euclidean error in three-dimensional space as illustrated in Figure 1. Our approach extends the traditional Lloyd–Max algorithm to efficiently compress point cloud data by applying it independently to the distribution of points along each of the three dimensions. This allows the resulting voxels to take on generalised cuboid shapes rather than being restricted to cubes. As shown in Equation (Equation 8), minimising the Euclidean error in three-dimensional space reduces to minimising the squared error along each dimension independently, assuming decision boundary planes that are orthogonal to the axes. Thus, our approach achieves minimal distortion within these constraints:(8)$$\begin{array}{cc} D_{X,Y,Z} & =E[d{\left(P,Q\left(P\right)\right)}^{2}]\\ \; & =\int_{-\infty }^{\infty }\int_{-\infty }^{\infty }\int_{-\infty }^{\infty }d{\left(P,Q\left(P\right)\right)}^{2}f\left(x,y,z\right)dxdydz\\ \; & =\sum_{k_{1}=1}^{N}\sum_{k_{2}=1}^{N}\sum_{k_{3}=1}^{N}\int_{b_{x,k_{1}-1}}^{b_{x,k_{1}}}\int_{b_{y,k_{2}-1}}^{b_{y,k_{2}}}\int_{b_{z,k_{3}-1}}^{b_{z,k_{3}}}{\sqrt{{\left(x-{\hat{x}}_{k_{1}}\right)}^{2}+{\left(y-{\hat{y}}_{k_{2}}\right)}^{2}+{\left(z-{\hat{z}}_{k_{3}}\right)}^{2}}}^{2}f\left(x,y,z\right)dxdydz\\ \; & =\sum_{k_{1}=1}^{N}\sum_{k_{2}=1}^{N}\sum_{k_{3}=1}^{N}\int_{b_{x,k_{1}-1}}^{b_{x,k_{1}}}\int_{b_{y,k_{2}-1}}^{b_{y,k_{2}}}\int_{b_{z,k_{3}-1}}^{b_{z,k_{3}}}{\left(x-{\hat{x}}_{k_{1}}\right)}^{2}f\left(x,y,z\right)dxdydz+\ldots \\ \; & =\sum_{k_{1}=1}^{N}\int_{b_{x,k_{1}-1}}^{b_{x,k_{1}}}\left(\sum_{k_{2}=1}^{N}\sum_{k_{3}=1}^{N}\int_{b_{y,k_{2}-1}}^{b_{y,k_{2}}}\int_{b_{z,k_{3}-1}}^{b_{z,k_{3}}}{\left(x-{\hat{x}}_{k_{1}}\right)}^{2}f\left(x,y,z\right)dydz\right)dx+\ldots \\ \; & =\sum_{k_{1}=1}^{N}\int_{b_{x,k_{1}-1}}^{b_{x,k_{1}}}{\left(x-{\hat{x}}_{k_{1}}\right)}^{2}f\left(x\right)dx+\ldots \\ \; & =D_{X}+D_{Y}+D_{Z}. \end{array}$$

While these assumptions may appear restrictive, they are crucial for the practical application of non-uniform quantisation to point clouds. Although a vector quantisation approach might initially appear more suitable due to its ability to perform generalised multi-dimensional quantisation, voxelisation necessitates the creation of an N×N×N grid, which vector quantisation is not designed for. Additionally, the reconstruction values must be included in the geometry header to ensure accurate point cloud reconstruction on the decoder side. Making the quantisers interdependent introduces a size complexity of $N^{3}$ for the header overhead, which is impractical for real-world applications. As we will demonstrate, even the 3N complexity of independent quantisers poses significant challenges. Finally, optimising and encoding non-planar quantisation boundaries is beyond the scope of this work.

Another important consideration is the execution time of the voxelisation process. While extracting the point distribution and determining an appropriate quantiser along each axis is straightforward and efficient, directly processing the points in three-dimensional space is far more complex. These complexities are a key reason why most existing approaches incorporate a voxelisation step in the first place.

To accurately reconstruct the point cloud at the decoder, it is necessary to encode the mapping between the internal integer coordinate system and the actual point cloud coordinates. However, the bitrate required is very substantial due to the large number of voxels along each dimension, making it impractical to transmit these data directly. As the octree depth increases, the number of reconstruction levels grows exponentially, with each level requiring a 32-bit float representation. At higher-quality levels, the overhead of transmitting this information becomes prohibitive. Figure 2 illustrates the considerable overhead introduced by communicating the reconstruction levels compared to just the outer boundaries of a uniform approach. The impact of the header overhead on the efficiency of the proposed method will be further explored in Section 4.3.

To address the overhead, our approach applies an additional Lloyd–Max quantiser to the reconstruction levels in order to reduce the introduced overhead. By quantising the output reconstruction levels of the Lloyd–Max quantiser, we obtain an approximate version of the original quantiser. The primary goal is to minimise the additional distortion, denoted as $\hat{D}$, introduced by this approximation, relative to the optimal distortion achieved by the original quantiser. It is important to note that only the reconstruction values need to be communicated and thus approximated, meaning both quantisers share the same boundaries:(9)$$\begin{array}{cc} \hat{D} & =E\left[{\left(x-\hat{Q}\left(x\right)\right)}^{2}\right]-E\left[{\left(x-Q\left(x\right)\right)}^{2}\right]\\ \; & =E\left[\hat{Q}{\left(x\right)}^{2}-Q{\left(x\right)}^{2}\right]-2E\left[x\hat{Q}\left(x\right)-xQ\left(x\right)\right]\\ \; & =\sum_{n=1}^{N}\left({\hat{y}}_{n}^{2}-y_{n}^{2}\right)\int_{b_{n-1}}^{b_{n}}f\left(x\right)dx-2\sum_{n=1}^{N}\left({\hat{y}}_{n}-y_{n}\right)\int_{b_{n-1}}^{b_{n}}xf\left(x\right)dx\\ \; & =\sum_{n=1}^{N}\left({\hat{y}}_{n}^{2}-y_{n}^{2}\right)p_{n}-2\sum_{n=1}^{N}\left({\hat{y}}_{n}-y_{n}\right)p_{n}y_{n}\\ \; & =\sum_{n=1}^{N}{\left({\hat{y}}_{n}-y_{n}\right)}^{2}p_{n}, \end{array}$$
where $\hat{Q}$ and ${\hat{y}}_{n}$ denote the approximated reconstruction values, and $p_{n}$ is the probability or importance of a certain interval. We apply quantisation on the reconstruction levels only after a prediction step for increased efficiency. Thus, we can rewrite the additional distortion in terms of the differences $d_{n}=y_{n}-y_{n-1}$:(10)$$\begin{array}{c} \hat{D}=\sum_{k=1}^{K}\sum_{\left\{n|b_{k-1}<d_{n}<b_{k}\right\}}{\left(d_{n}-\hat{d_{k}}\right)}^{2}p_{n}, \end{array}$$
which is a discrete version of the classical distortion (Equation (Equation 2)). So we apply the second Lloyd–Max quantiser to the differences $d_{n}=y_{n}-y_{n-1}$, using the importance $p_{n}$ for the calculation of the mean value for each bin (Equation (Equation 7)).

## 4. Results

### 4.1. Datasets

The efficiency of the proposed method is demonstrated through performance evaluation on three widely used public datasets. ScanNet [26] is a popular 3D indoor scene point cloud dataset commonly used for tasks such as segmentation, compression, and other point cloud processing applications. We evaluate the proposed method using 20 scenes from this dataset. The other datasets, ModelNet [29] and ShapeNet [28], consist of synthetic meshes representing various object categories. For evaluation, we select two meshes from ten categories in both datasets, sampling 1,000,000 points and their associated normals from each mesh.

### 4.2. Experimental Details

All voxelisation in the state-of-the-art point cloud compression research is uniform, which thus serves as the baseline for comparison for our proposed non-uniform voxelisation approach. To measure the effect of the proposed method on bitrate, we encode the voxelised point cloud using the G-PCC standard [4], which is a state-of-the-art point cloud compression codec that is widely used as a benchmark in the point cloud compression field. Rate-distortion tuning is achieved by adjusting the octree depth parameter, which in our experiments varies between 6 and 15, giving us a wide range of voxelised point cloud densities. To evaluate performance, we use four key metrics. The bitrate is measured in bits per point (bpp). The reconstruction quality is assessed by calculating the PSNR of the point-to-point and point-to-plane distortions as defined in [30], with p=1. Lastly, the Bjontegaard Delta rate (BD-rate) is employed to quantify the rate-distortion gains; it measures the bitrate difference for the same reconstruction quality. All experiments are performed on a desktop system with an Intel Core i9-9900K CPU and 64 GB of memory. A new parameter introduced by the proposed method is the number of bits used for the reconstruction values in the second quantisation, referred to as Q-bits. Lower values result in significant rate savings, while higher values yield higher-quality reconstructions. Empirical results indicate that setting this parameter to four provides optimal performance as detailed in Section 4.3.

### 4.3. Q-Bits Tuning

In addition to the octree depth, which defines the number of quantisation bits for the first quantiser, the proposed approach introduces a second tunable parameter, the number of quantisation bits for the second quantiser, referred to as Q-bits. We investigate the effects of tuning this parameter and identify the optimal values for performance. The results are shown in Table 1. It can be observed that a Q-bit value of four performs well across the different datasets and distortion metrics. Another key observation is the significance of the second quantisation, as the method’s performance begins to degrade significantly at 12 bits per reconstruction value. Furthermore, the method exhibits robust performance across a range of smaller Q-bit values, suggesting that fine-tuning this parameter is not required to achieve good results.

### 4.4. Comparison to SOTA

The quantitative results of the comparison between the proposed method and uniform voxelisation are presented in Table 2 and Figure 3. These results demonstrate the efficacy of the proposed method, with average BD-rate gains of 9.4% and 27.91% for point-to-point and point-to-plane distortion, respectively. The proposed method consistently achieves significantly higher PSNRs while incurring only a slight increase in rate, showing that it can effectively minimise 3D distortion while the second quantiser reduces the bitrate overhead sufficiently. A qualitative comparison between the proposed method and the baseline is shown in Figure 4. The strength of the non-uniform approach lies in its ability to adapt to the point distribution within the input point cloud, such as aligning the reconstruction values with dense regions. This is clearly demonstrated in the examples, particularly in the seat of the ModelNet chair, specific surfaces of the ShapeNet bunk bed, and the floor and walls in the ScanNet scene. In contrast, the uniform approach fails to account for point distribution, often resulting in large errors in these critical regions.

### 4.5. Impact of Density

Point cloud compression methods are designed to handle a wide range of densities, from sparse point clouds captured by real-time depth sensors to highly dense point clouds used in applications such as heritage preservation. To assess the impact of point cloud density on the performance of the proposed method, we conducted experiments using the ModelNet dataset. Since ModelNet provides mesh data, we sampled it to generate point clouds with varying densities, allowing for controlled experimentation across a broad range of point counts. Specifically, we sampled point clouds containing between 50,000 and 4,000,000 points, which reflects the typical density variations encountered in real-world datasets as summarised in Table 3.

The results of the density experiment are presented in Table 4. The findings indicate that the proposed method performs better for denser point clouds. This can be attributed to the reduced relative overhead of communicating reconstruction values, which does not scale proportionally with the number of points. The proposed method consistently outperforms the uniform voxelisation approach across all density levels. Additionally, as the density increases, the rate allocated to quantisation becomes a smaller fraction of the total rate, enabling higher Q-bits values without a significant increase in cost. These results highlight the robustness and scalability of the proposed method, making it particularly well suited for applications that involve high-density point clouds.

### 4.6. Execution Time

The final aspect of the proposed method to evaluate is its execution time. While the method’s adaptability to the input point cloud enhances its performance, it also introduces additional computational complexity. This is due to the iterative nature of the Lloyd–Max quantiser, with each iteration having a complexity of O(inputsize) [35]. The execution times are provided in Table 5. Despite this increased complexity, the rise in execution time is relatively small compared to the overall processing time of the G-PCC codec. Therefore, the trade-off in execution time is modest when weighed against the significant improvements in coding efficiency offered by the proposed method.

## 5. Discussion

The experimental results clearly demonstrate that the proposed method achieves significant improvements in coding performance compared to uniform voxelisation, which is predominantly used by state-of-the-art geometry-based point cloud compression codecs. These improvements stem from the method’s ability to adapt voxelisation to the distribution of the input point cloud. While the method’s performance is particularly effective with denser point clouds and higher bitrates, it also shows robust results across a wide range of practical scenarios. This robustness is achieved with minimal additional tuning and only a minor increase in execution time. Furthermore, the method can be seamlessly integrated with existing state-of-the-art techniques that operate on voxelised point clouds, enhancing its practical applicability. Despite these strengths, the approach has some limitations. Its reliance on orthogonal quantisation boundaries limits the theoretically possible reconstruction quality. Addressing this limitation while maintaining reasonable bitrates and execution times could lead to further performance gains. Additionally, exploring the integration of deep learning algorithms presents an exciting opportunity to overcome these constraints. The adaptive nature of deep learning techniques could enhance the method’s ability to refine voxelisation and compression performance, potentially unlocking new capabilities in point cloud compression.

## 6. Conclusions

This paper presents a novel approach to point cloud compression by introducing non-uniform, distortion-minimising voxelisation, representing the first exploration of non-uniform quantisation in this domain. Unlike traditional methods that rely on uniform voxelisation, the proposed technique adapts voxel sizes to the local distribution of points, thereby optimising compression performance. The method’s efficiency and robustness were thoroughly validated through extensive experiments on the well-established public datasets ScanNet, ModelNet and ShapeNet. The results highlight the potential for the further exploration of innovative non-uniform voxelisation techniques to advance the field of point cloud compression.

## Figures and Tables

**Figure 1 sensors-25-00865-f001:**
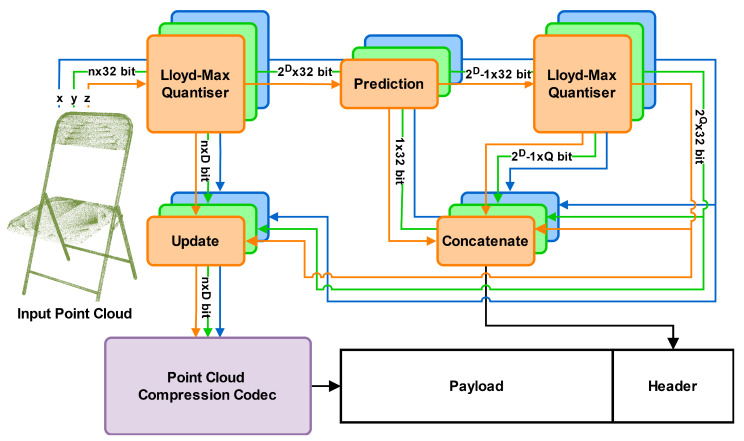
Overview of the proposed method. An input-specific voxelisation is decided by applying Lloyd–Max quantisation for each of the input dimensions. Subsequently, another round of Lloyd–Max quantisation is applied to the reconstruction values after a linear prediction in order to reduce their overhead dramatically. Finally, the input point cloud is voxelised using the determined quantisers, ready for compression using any voxel- or octree-based codec. The quantiser information, which is necessary for the calculation of reconstruction values, is stored in the header. The different colours signify the independent processing, and all denoted bitrate sizes are for a singular dimension.

**Figure 2 sensors-25-00865-f002:**
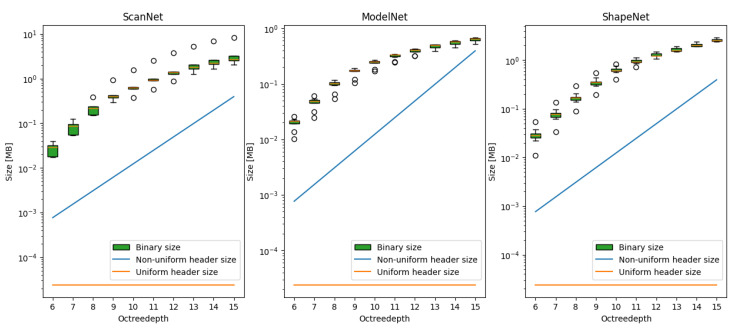
Illustration of the large bitrate overhead of a naive non-uniform approach for the ScanNet, ModelNet, and ShapeNet datasets. The increased overhead would quickly nullify the benefit of any increase in compression performance. The point clouds are compressed using uniform voxelisation and G-PCC. The overhead costs are 3×2×4B and $3\times 2^{octreedepth}\times 4B$ for the uniform and non-uniform approaches, respectively.

**Figure 3 sensors-25-00865-f003:**
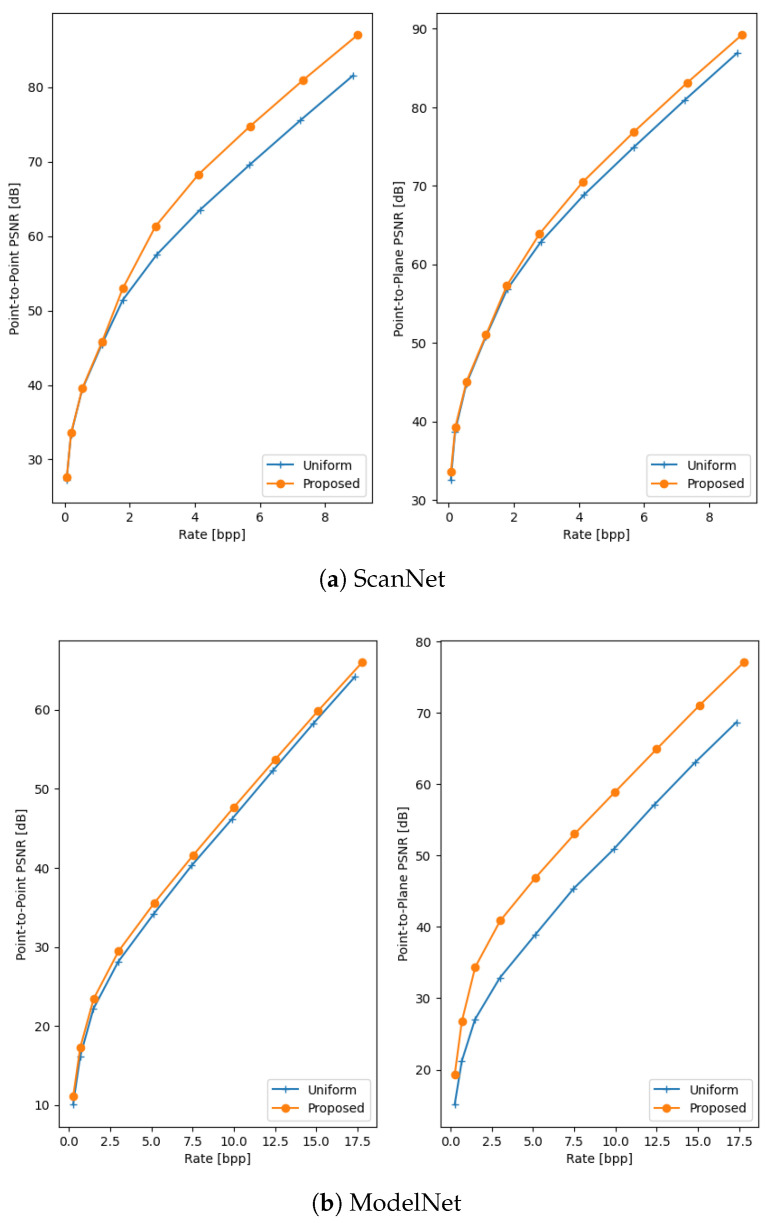
Rate-Distorion curves of the proposed method and the baseline on the ScanNet (**a**), ModelNet (**b**) and ShapeNet (**c**) datasets.

**Figure 4 sensors-25-00865-f004:**
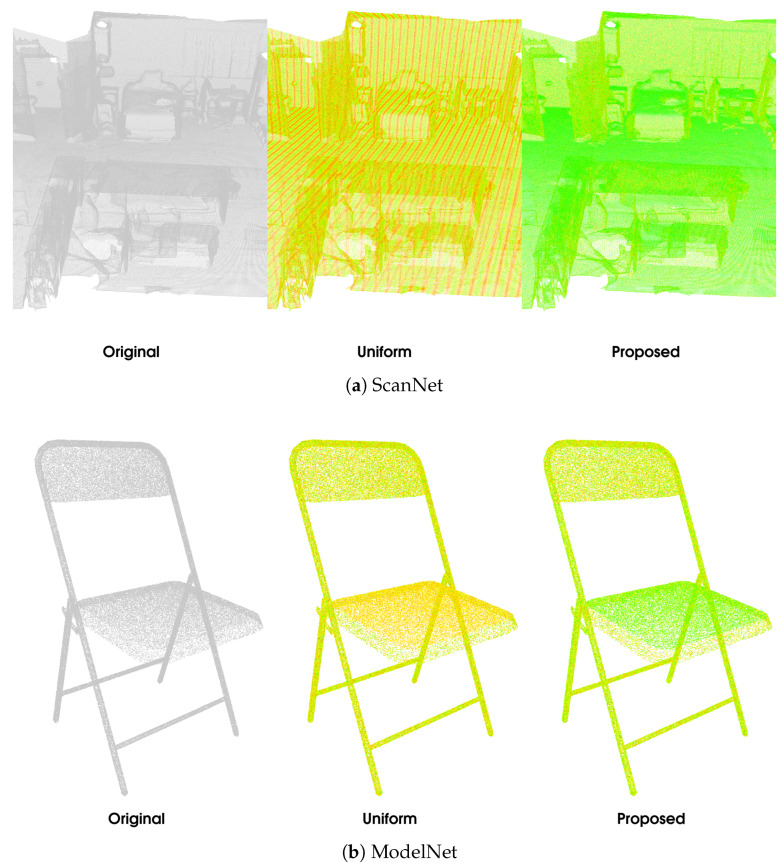
A qualitative comparison of uniform voxelisation and the proposed non-uniform voxelisation on a sample from the ScanNet (**a**), ModelNet (**b**), and ShapeNet (**c**) datasets. Each point is coloured from red to green based on the Euclidian coding error (D1) for that point; the colour scale is shared between the methods.

**Table 1 sensors-25-00865-t001:** Quantitative comparison of the coding performance for different Q-bits values, the metric being BD-rate gains (point-to-point and point-to-plane) compared against the uniform approach. The best results are ***highlighted***.

Q-Bits	ScanNet		ModelNet		ShapeNet	
	D1 BD-Rate	D2 BD-Rate	D1 BD-Rate	D2 BD-Rate	D1 BD-Rate	D2 BD-Rate
2	−11.11%	−5.51%	−5.66%	−25.05%	−5.56%	−24.56%
3	−12.04%	−6.35%	−8.20%	−38.20%	−8.02%	−39.30%
4	* **−12.21%** *	* **−6.51%** *	* **−8.40%** *	−41.49%	* **−8.23%** *	−46.14%
6	−11.91%	−6.15%	−8.03%	* **−42.49%** *	−7.81%	−50.29%
8	−10.76%	−4.90%	−6.82%	−41.71%	−6.52%	* **−50.40%** *
12	5.92%	12.61%	9.90%	−28.94%	11.19%	−38.06%

**Table 2 sensors-25-00865-t002:** Comparison of rate-distortion performance of the proposed method using the BD-rate gain. D1 and D2 refer to point-to-point and point-to-plane distortions, respectively.

Data		Uniform			Proposed			BD-Rate Gain	
		Rate	D1 PSNR	D2 PSNR	Rate	D1 PSNR	D2 PSNR	D1	D2
Scannet		3.25	54.45	59.80	3.27	57.17	60.97	−12.21	−6.51
ModelNet	bathtub	7.52	35.75	39.41	7.66	37.72	46.64	−11.94	−40.23
	bed	7.17	32.84	38.22	7.26	33.82	42.49	−6.13	−29.17
	chair	5.69	40.26	44.76	5.78	41.24	47.58	−7.20	−23.39
	desk	8.51	33.92	38.69	8.60	35.58	50.07	−8.86	−47.41
	dresser	7.84	35.01	39.13	7.94	37.05	54.98	−11.19	−59.42
	monitor	5.68	44.33	49.48	5.81	45.91	61.59	−10.33	−62.53
	nightstand	7.71	38.33	42.34	7.81	39.83	47.73	−8.77	−31.09
	sofa	7.36	34.09	42.25	7.45	34.53	45.04	−1.90	−19.69
	table	6.95	36.18	40.57	7.14	37.90	49.79	−10.67	−49.41
	toilet	7.95	41.08	45.34	8.04	41.82	47.44	−5.11	−16.61
ShapeNet	airplane	6.08	74.09	81.09	6.16	74.52	82.53	−1.45	−11.55
	trash bin	8.44	73.79	78.46	8.53	74.45	80.43	−4.02	−13.42
	bag	6.82	73.89	80.52	6.90	74.43	83.07	−3.49	−19.94
	basket	7.13	73.34	77.22	7.11	75.35	88.54	−13.99	−58.37
	bathtub	7.92	73.85	79.12	8.00	74.25	80.19	−1.81	−7.40
	bed	6.72	73.49	77.36	6.87	75.77	93.39	−14.25	−68.60
	bench	6.54	73.50	79.13	6.62	74.52	94.55	−6.70	−66.25
	birdhouse	6.73	73.46	77.89	6.91	75.06	85.23	−9.00	−41.44
	bookshelf	7.96	73.50	76.93	8.14	76.18	97.76	−14.84	−71.49
	bottle	7.37	73.49	78.75	7.46	74.69	84.02	−8.23	−34.80
Average		5.89	55.09	60.16	5.96	56.88	65.76	−9.40	−27.91

**Table 3 sensors-25-00865-t003:** Densities of commonly used datasets for point cloud processing.

Dataset	Average Points per Frame
ScanNet [26]	3.3 M
KITTI [31]	120 K
nuScenes [32]	34 K
Waymo Open Dataset [33]	177 K
S3DIS [34] (Room/Area)	1 M/45 M

**Table 4 sensors-25-00865-t004:** Performance on point clouds with different densities.

# Points	Q-Bits	Uniform			Proposed			BD-Rate Gain	
		Rate	D1 PSNR	D2 PSNR	Rate	D1 PSNR	D2 PSNR	D1	D2
50,000	2	12.60	37.18	42.00	13.46	39.03	47.62	−3.31	−19.55
100,000	3	11.18	37.18	42.00	11.83	38.97	48.85	−4.76	−27.32
250,000	4	9.49	37.19	42.04	9.85	38.74	49.39	−6.16	−34.21
500,000	4	8.30	37.18	42.02	8.50	38.61	49.37	−7.40	−38.05
1,000,000	4	7.24	37.18	42.02	7.35	38.54	49.33	−8.40	−41.49
4,000,000	5	5.38	37.19	42.04	5.42	38.49	49.58	−10.51	−49.78

**Table 5 sensors-25-00865-t005:** Comparison between the execution time of the complete codec using uniform and non-uniform voxelisation. The runtimes are expressed in milliseconds.

Octree Depth	ScanNet			ModelNet			ShapeNet		
	Uniform	Proposed	Δ	Uniform	Proposed	Δ	Uniform	Proposed	Δ
6	5194.60	5613.99	8.07%	1804.75	2180.99	20.85%	3253.65	3498.38	7.52%
7	5190.60	5508.96	6.13%	1867.25	2059.26	10.28%	3219.40	3548.07	10.21%
8	5143.20	5543.84	7.79%	1909.05	2090.62	9.51%	2930.35	3481.01	18.79%
9	5257.00	5678.42	8.02%	2111.20	2280.34	8.01%	3404.70	3491.80	2.56%
10	5759.20	6108.41	6.06%	2354.25	2513.61	6.77%	3849.10	4067.62	5.68%
11	6593.20	6862.53	4.09%	2603.80	2729.44	4.83%	3977.65	4079.22	2.55%
12	7177.60	7666.90	6.82%	2768.40	2966.85	7.17%	4059.70	4341.99	6.95%
13	7903.00	8410.00	6.42%	2989.00	3183.79	6.52%	4512.95	4649.83	3.03%
14	8644.40	9112.04	5.41%	3161.90	3413.44	7.96%	4807.10	5077.17	5.62%
15	9239.60	9910.39	7.26%	3454.80	3665.64	6.10%	5201.80	5459.60	4.96%

## Data Availability

The ScanNet dataset is available at http://www.scan-net.org/ (accessed on 5 October 2021), the ModelNet dataset is available at https://modelnet.cs.princeton.edu/ (accessed on 7 August 2024) and the ShapeNet dataset is available at https://shapenet.org/ (accessed on 17 January 2025.

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
