# Peer review of "Non-Uniform Voxelisation for Point Cloud Compression"

_sensors, 2025, doi:10.3390/s25030865_

Round 1

Reviewer 1 Report

Comments and Suggestions for Authors

This paper proposes a non-uniform voxelisation method for point cloud compression based on Lloyd-Max quantisation in each axis and an additional quantisation step for reconstruction values. The experimental evaluation shows improved rate-distortion performance over uniform voxelisation, especially for dense point clouds. The paper is clearly written, though certain methodological and reporting details could be clarified or expanded.

 Comments

1. Please provide more details on how the second Lloyd-Max quantiser is tuned and integrated, such as how exactly the reconstruction differences (dn) are collected and quantised, to ensure reproducibility.  

2. In Section 3, the claim that minimizing 3D Euclidean error reduces to summing three separate 1D distortions is briefly stated; a clearer derivation or reference would strengthen the justification of your approach.  

3. In Figure 2, it is unclear how you compute the overhead cost of encoding reconstruction values; please clarify the bit budget assumptions for both uniform and non-uniform cases.  

4. In Table 2, clarify why the rate for the proposed method slightly exceeds the uniform case in some rows yet still yields overall BD-rate savings; a short discussion on the trade-off between bit overhead and distortion is needed.  

5. The paper mentions computing run times (Table 5) but lacks explanations on the computational cost of the Lloyd-Max iterations; providing a short complexity or iteration count analysis would help readers assess real-world feasibility.

Reviewer 2 Report

Comments and Suggestions for Authors

The paper proposes a novel and promising approach, and its advantages are demonstrated through experimental validation. However, there are areas that require further clarification and improvement, particularly regarding algorithmic details, experimental design, and practical applicability. It is recommended that the authors address the following comments to enhance the quality and depth of the manuscript:

1. The authors mention using the ScanNet and ModelNet datasets for experiments. Have other datasets been considered to further validate the performance of the proposed method? For instance, datasets such as ShapeNet may provide diverse experimental results across different tasks and scenarios.

2. In the experimental section, the comparison appears to be limited to the uniform approach. It would strengthen the study to include comparisons with other methods to provide a more comprehensive evaluation.

3. In line 188, page 7, the authors state that the octree depth in the experiments varies between 6 and 16. Could the authors clarify the rationale behind selecting this specific range?

4. Table 2 presents examples of multiple objects; however, only two examples are listed in Figure 4. It is suggested that the authors elaborate on why these specific examples were chosen and whether additional examples could further substantiate their findings.

5. In line 193, page 7, the authors provide CPU and memory information but do not include details on the graphics card used. Since the graphics card may impact processing time, it would be helpful if the authors could elaborate on this aspect.

6. A discussion section is recommended to better contextualize the proposed method by analyzing its differences from other approaches in the literature. This could provide valuable insights into the novelty and potential impact of the research.

Round 2

Reviewer 1 Report

Comments and Suggestions for Authors

The authors have addressed the comments properly.

Reviewer 2 Report

Comments and Suggestions for Authors

The author's revisions are satisfactory, and it is recommended that the journal accept the paper.